# Are Dietary Indices Associated with Polycystic Ovary Syndrome and Its Phenotypes? A Preliminary Study

**DOI:** 10.3390/nu13020313

**Published:** 2021-01-22

**Authors:** Ana Cutillas-Tolín, Julián Jesús Arense-Gonzalo, Jaime Mendiola, Evdochia Adoamnei, Fuensanta Navarro-Lafuente, María Luisa Sánchez-Ferrer, María Teresa Prieto-Sánchez, Ana Carmona-Barnosi, Jesús Vioque, Alberto M. Torres-Cantero

**Affiliations:** 1Division of Preventive Medicine and Public Health, Department of Public Health Sciences, University of Murcia School of Medicine, 30100 Espinardo Murcia, Spain; julianjesus.arense@um.es (J.J.A.-G.); jaime.mendiola@um.es (J.M.); fuensanta.navarro1@um.es (F.N.-L.); amtorres@um.es (A.M.T.-C.); 2Institute for Biomedical Research of Murcia, IMIB-Arrixaca, 30120 El Palmar Murcia, Spain; evdochia.adoamnei@um.es (E.A.); marisasanchez@um.es (M.L.S.-F.); mt.prieto@um.es (M.T.P.-S.); 3Department of Nursing, University of Murcia School of Nursing, 30100 Espinardo Murcia, Spain; a.carmonabarnosi@um.es; 4Department of Obstetrics and Gynecology, Virgen de la Arrixaca University Clinical Hospital, 30120 El Palmar Murcia, Spain; 5Consortium for Biomedical Research in Epidemiology and Public Health (CIBER Epidemiología y Salud Pública, CIBERESP), Instituto de Salud Carlos III, 28029 Madrid, Spain; vioque@umh.es; 6Nutritional Epidemiology Unit, ISABAL-Miguel Hernandez University, 03550 San Juan de Alicante, Spain; 7Department of Preventive Medicine, University Clinical Hospital Virgen de la Arrixaca, 30120 El Palmar Murcia, Spain

**Keywords:** polycystic ovary syndrome (PCOS), PCOS phenotypes, Rotterdam criteria, hyperandrogenism, Mediterranean diet score (MDS), alternate healthy index (AHEI), alternate healthy index 2010 (AHEI-2010) and DASH index

## Abstract

Polycystic ovary syndrome (PCOS) is a complex hormonal disorder which impairs ovarian function. The adherence to healthy dietary patterns and physical exercise are the first line of recommended treatment for PCOS patients, but it is yet unclear what type of diet is more adequate. In this case-control study, we explored associations between adherence to five dietary quality indices and the presence of PCOS. We enrolled 126 cases of PCOS and 159 controls living in Murcia (Spain). Diagnostic of PCOS and its phenotypes were established following the Rotterdam criteria (hyperandrogenism (H), oligoanovulation (O), polycystic ovaries morphology (POM)). We used a validated food frequency questionnaires to calculate the scores of five dietary indices: alternate Healthy Eating index (AHEI), AHEI-2010, relative Mediterranean Dietary Score (rMED), alternate Mediterranean Dietary Score (aMED) and Dietary Approaches to Stop Hypertension (DASH). We used multivariable logistic regression to estimate adjusted odds ratios and confidence intervals. In the multivariable analysis, AHEI-2010 index was inversely associated with Hyperandrogenism + Oligoanovulation PCOS phenotype (OR_Q3_ vs. _Q1_ = 0.1; 95% CI: (0.0; 0.9); P_for trend_ = 0.02). We did not find any statistical significant association between dietary indices and total anovulatory or ovulatory PCOS. However, further studies with higher sample sizes exploring these associations among the diverse phenotypes of PCOS are highly warranted.

## 1. Introduction

Polycystic ovary syndrome (PCOS) is a complex hormonal disorder which impairs ovarian function [1]. It could be considered a polygenic, polyfactorial, systemic and inflammatory disease [2]. Its reported overall prevalence according to the Rotterdam criteria is about 10% (95% CI: 8–13%) (Skiba et al., 2018). The impact of PCOS is considerable because it is linked to higher risk of obesity [3,4,5,6], insulin resistance and diabetes [7,8,9], higher cardiovascular risk profile [10,11,12], poor thyroid function [13,14], infertility [15,16], gestational diabetes [17,18,19,20], sleep disturbances [21] and even mental health problems [22].

Lifestyle modifications for PCOS, especially physical activity and diet, are of major importance in the clinical management of women to improve most of the adverse outcomes related to this condition. Diet is focused on weight loss in overweight PCOS women, the subgroup with higher risk of metabolic deregulation and type 2 diabetes. However, it is unclear what type of diet is better for this: hypocaloric or low in carbohydrates.

On the one hand, some case-control studies have shown that PCOS women consumed higher quantity of monounsaturated fatty acids, and ω-3 polyunsaturated fatty acid and simple carbohydrate (Barrea et al., 2019). Moreover, they presented higher fiber intake, high glycemic index and glycemic load than controls [23,24]. All these key aspects are considered in different dietary indices such as the alternate Mediterranean Diet Score (aMED), relative Mediterranean Diet Score (rMED), Alternate Healthy Index (AHEI) and Dietary Approaches to Stop Hypertension (DASH).

On the other hand, most interventional studies have employed low carbohydrates and high protein diets for improving different PCOS manifestations [25,26,27,28,29,30]. A recent meta-analysis of eight randomized controlled trials concluded that a low carbohydrate diet had a stronger effect on increasing FSH level, rising SHBG levels and decreasing testosterone levels comparing with a higher carbohydrate diet (diet composition carbohydrates: 40% vs. 50%) [25]. However long-term adherence may be difficult. Moreover, an unanswered question is to what extent improvements will be maintained in association with a transition to a less carbohydrate-restricted diet. Indeed, low carbohydrate diet interventions do not often compare with other frequent healthy dietary patterns such as the Mediterranean or DASH diet as the control group. In this way, DASH interventions for PCOS women have had beneficial effects on body mass index (BMI), androstenedione, sex hormone-binding globulin (SHBG), insulin metabolism, cardiovascular risk factors and oxidative stress [31,32,33,34].

Because of that, our aim was to evaluate if there are differences in dietary indices in women diagnosed with PCOS and its phenotypes compared to controls. To our knowledge, this is the first study evaluating associations between Mediterranean Diet indices (aMED and rMED) and different phenotypes of PCOS. Additionally, AHEI, Alternate Healthy Index 2010 (AHEI-2010) and DASH were also assessed. This approach would allow us to know if diet patterns may be related to PCOS, and not only specific macro or micronutrients. Besides, we would be able to report more easily nutritional recommendations potentially adapted to PCOS women.

## 2. Materials and Methods

This was a case-control study taken place in Southeast of Spain (Murcia Region) from September 2014 to May 2016. All participants were between 18 and 40 years old (*n* = 300). Exclusion criteria were pregnancy or lactating, oncological treatment, hormonal medication during the three months prior to the study, genitourinary prolapse or endocrine disorders (*n* = 5). Methods were explained in previous works [35,36,37,38]. Figure 1 shows the number of participants and the exclusion criteria. Concisely, we finally enrolled, voluntarily, 126 cases of PCOS and 159 women from the Department of Obstetrics and Gynecology, outpatient clinics at the Virgen de la Arrixaca University Clinical Hospital. We excluded, because of the objective of this paper, five subjects who did not complete the questionnaire about food intake and four with an implausible total kcal per day. Thus, the analysed sample size were 276, being 121 PCOS cases and 155 control. In a second phase, we carried out sensitivity tests excluding women:taking medication that interferes carbohydrate metabolism, not excluded previously (cases = 100; controls = 134)taking multivitamins 10–12 months (cases = 98; controls = 131)taking multivitamins 4–6 months or 10–12 months (cases = 90; controls = 123)

Diagnostic of PCOS were established following the Rotterdam criteria resulting from the consensus of European Society for Human Reproduction and Embryology and the American Society for Reproductive Medicine [39]. Thus, PCOS was diagnosed with two or more of the following criteria:oligovulation/amenorrhea or anovulation (menstrual cycles > 35 days or amenorrhea > 3 months).biochemical hyperandrogenism (total testosterone level ≥ 2.6 nmol or clinical (Ferriman-Galwey score ≥ 12) [40].polycystic ovaries morphology (POM) using transvaginal ultrasound (TVUS) (≥12 follicles measuring 2–9 mm in diameter, mean of both ovaries) [41].

Moreover, the following phenotypes of PCOS were also assessed [42]:hyperandrogenism + oligo/amenorrhea + POM (H + O + POM) (*n* = 52).hyperandrogenism + oligo/amenorrhea (H + O) (*n* = 18).hyperandrogenism + POM (H + POM) (*n* = 33).oligo/amenorrhea + POM (O + POM) (*n* = 18).

Also, H + O + POM, H + O and O + POM phenotypes were reclassified, as anovulatory phenotype” (*n* = 88) and H + POM type as ovulatory phenotype (*n* = 33) and evaluated separately in the current study.

Controls were women without PCOS (or other major gynecological conditions) attending the gynecological outpatient clinic for routine gynecological examinations. The same methods were performed in both cases and controls: anamnesis and questionnaires, physical examination, transvaginal ultrasound and blood draw, between days 2–5 of the menstrual cycle. Written informed consent was obtained from all women. This study was approved by the Ethics Research Committee of the University of Murcia and the Clinical University Hospital Virgen de la Arrixaca (no. 770/2013, approved 3 October 2013).

### 2.1. Dietary Assessment and Dietary Indices

We used a validated 117-food item semiquantitative food frequency questionnaire (FFQ) to assess the regular food intake which was previously validated for the Spanish population [43,44]. This questionnaire is based on a FFQ used by Willett and collaborators in the Nurse Health Study Cohort [45]. Subjects had to choose one of the nine options about how often, on average, they had consumed each food ítem (from never or less than once a month, to six or more times a day). Nutrient values for each food were obtained from the United States Department of Agriculture and supplemented with Spanish sources [46,47].

The FFQ dietary information was used to calculate the following five a priori-defined dietary indices:AHEIAHEI-2010rMEDaMEDDASH.

All these represent healthy dietary patterns but use different range of scores, variations in food components and calculations. They are described in detail in a previous publication [48]. AHEI, AHEI-2010 and DASH were created in the United States to define a prudent dietary pattern high in vegetables, fruit, whole grains, legumes and lower in saturated fats and alcohol [49,50,51]. The AHEI was developed by Teresa Fung and coworkers to quantify adherence to the US federal dietary guidance of 1992, with a higher score reflecting better quality and adherence. The scores range from 0 to 87.5. The score evaluates nine components such as trans fats, protein sources, polyunsaturated/monounsaturated ratio and cereal fiber. Table 1 describes the dietary indices in detail. AHEI-2010 is the AHEI’s version for evaluating chronic diseases. AHEI and AHEI-2010 establish specific reference values of servings per day or grams per day for each food component and sum 10 if the subject reaches this amount. AHEI-2010 was designed in 2012 by Chiuve and colleagues based on updated literature to study the relation between food intake and chronic diseases. The AHEI-2010 scores 11 components for a total of 110 points, including whole grains intake (specific for women), legumes and nuts, red/processed meat ratio, sugar-sweetened beverages and fruit juices, sodium and polyunsaturated fats [52]. The overall scoring range is 0 to 80 for AHEI and 0 to 110 for AHEI-2010. However, DASH, rMED and aMED establish the scoring criteria using quintiles, terciles and the median intake of the study sample, respectively. DASH was developed for controlling blood pressure [53] but, nowadays, is useful for obesity, diabetes, metabolic syndrome and cardiovascular disease. The DASH dietary pattern is rich in fruit, vegetables and low-fat dairy products. rMED and aMED indices define the Mediterranean Diet and are versions of the original Mediterranean Diet score [54,55]. aMED considers red and processed meat and establishes the ratio of mono/polyunsaturated fats [49], while rMED evaluates dairy products, only uses olive oil as the primary fat source, evaluates in one item all types of meat and is more specific for the Spanish population [56]. The total score is 9 and 18, respectively. In all of these indices, higher intakes of healthy food items such as fruits, vegetables, whole grains, nuts and legumes add higher scores, whereas higher intake of trans fats, meat, saturated fats, sodium and alcohol are associated with lower scores.

Energy-adjusted intakes were computed using the residual method [57]. After dietary assessment, we excluded five subjects who did not complete the FFQ and four did not have a plausible energy intake (≤500 or ≥4500 kcal) as shown in Figure 1. The study sample for statistical analyses was 276 women, being 121 cases and 155 controls.

### 2.2. Statistical Analyses

Data were checked for normal distributions using the Kolmogorov-Smirnov test. Continuous data with skewed distribution were described with median and interquartile ranges (IQR: 25th–75th) and comparisons were performed by the Kruskal-Wallis tests. We used the Chi-squared test for categorical variables and they were represented by frequency and percentage. The five dietary indices were recategorized in quartiles, the lowest quartile being the reference group.

We considered several variables as potential confounders and covariates (e.g., energy intake, nutrients intakes, physical activity, anthropometrics variables, age, gynaecological history). When inclusion of a potential covariate resulted in a change of the *p*-value corresponding to the dietary index variable less than 0.10, this covariate was kept in the final models. Hence, the covariates contained in the final models were total energy intake (kcal/day), body mass index (BMI) (Kg/m^2^), moderate-vigorous exercise (h/week), adjusted caffeine intake (mg/day) and adjusted carbohydrates intake (g/day). We used logistic regression to analyse the association between dietary indices (quartiles) and presence of PCOS, as well as PCOS phenotypes.

Additionally, we carried out a sensitivity analysis for evaluating if there was influence of medication interfering in the carbohydrate metabolism in our results. Concretely, we excluded from the analyses fifteen women taking corticosteroids (prednisone, betamethasone and gentamicin, deflazacort, budesonide, methylprednisolone), fourteen women taking hormonal contraceptives (emergency contraception, ethinyl drospirenone-estradiol, medroxyprogesterone acetate) and eight women taking thyroid hormones (*n* = 32). The final sample size was 232 subjects, 104 PCOS cases and 131 controls. In another sensitivity analysis we excluded women taking multivitamins and other s such us minerals, brewer’s yeast and probiotics. The prevalence was 9% (*n* = 21; total = 234) if we considered the use of vitamins during 4–6 months or 10–12 months in the previous year to the subject’s recruitment, and 2.1% (*n* = 5; total = 234) if we only considered use of vitamins during 10–12 months.

*p*-values ≤ 0.05 were considered statistically significant. All statistical analyses were performed in IBM SPSS 25.0 (IBM Corporation, Armonk, NY, USA).

## 3. Results

The average age was 29.1 (SD: 5.7) years. Table 2 shows demographic characteristics, metabolic parameters, hormonals determinations and nutrient intake across quartiles of adherence to healthful dietary scores for the study sample. AHEI, AHEI-2010, aMED indices presented a similar pattern. Women with higher adherence to any of these dietary scores had a higher age, physical activity and caffeine, carbohydrates and ω-3 fatty acids intake, but lower BMI. There was an increase of total energy intake across quartiles for AHEI, aMED and DASH scores, a decrease for rMED and no differences for AHEI-2010. In addition, greater adherence to any of these scores was associated with less intake of saturated fatty acids, except for the DASH score. Differences between median value scores across quartiles of rMED scores were different compared to the other dietary indices. Lastly, women with greatest adherences to rMED presented lower total energy and ω-3 fatty acids intake and were less physically active (Table 2). Differences in demographic characteristics, metabolic parameters and hormonal determinations among PCOS cases and controls have been published in a previous work [35].

In the multivariable analysis, we did not find any associations between the diet scores and total PCOS, neither in ovulatory nor anovulatory PCOS (Table 3). In the analyses by PCOS phenotypes, women with higher adherences to the AHEI-2010 pattern were less likely to present PCOS, specifically “H + O” phenotype (*p*-value for trend = 0.02) (Table 4). This association was mainly driven by vegetables food items of the AHEI-2010 index, being marginally significant (*p*-value for trend = 0.081). In contrast, we observed an inverse lineal trend between DASH index and “O + POM” phenotype (*p*-value for trend = 0.05).

Results from the four sensitivity analyses showed similar findings to those already presented, not changing the observed results.

## 4. Discussion

We found no clear associations between dietary indices and the presence of PCOS or its phenotypes. Only suggestive associations between some a priori-defined dietary indices (AHEI-2010 and DASH) and some PCOS phenotypes (H + O and O + POM, respectively) were observed.

In detail, women with higher adherences to the AHEI-2010 score were less likely to present H + O PCOS phenotype. At this point in time, there are only two studies evaluating the association between AHEI and PCOS. In the first study, no differences were detected in diet quality (using the Alternate Healthy Index-2015) or dietary intake between PCOS women and controls in a sample of women from New York (USA) [58]. However, in the second study, the Brazilian Healthy Eating Index was negatively correlated with BMI and waist circumference among PCOS patients [59].

We found that the Mediterranean diet was a protective factor only for the H + O phenotype. However, we must be cautious with this result since we did not consider subgrouping by PCOS phenotypes for the calculation of the study sample size. Thus, it is important to bear in mind this potential underpower. Nonetheless, we thought it was worthwhile exploring association among the diverse phenotypes of PCOS because diet may be more beneficial for women with a particular phenotype than for others.

One possibility which may explain this association is that women with the H + O phenotype presented higher prevalence of overweight and obesity (52.9%) compared to controls (33.1%) and other PCOS phenotypes (33.9%), but we need further studies with higher sample size. Indeed, nutritional randomized controlled trials in PCOS women have shown more effectiveness in reducing symptoms in women with overweight or obesity. Other reason could be that these randomized controlled trials, through diet, accomplished a reduction in hyperinsulinemia and, as a consequence, diminished hyperandrogenism. In previous studies, PCOS with hyperandrogenism increased around two folds (OR = 2.2; 95% CI: 1.9–2.6) the incidence of metabolic syndrome and three times (OR = 3.1; 95%CI: 2.3–4.2) the incidence of insulin resistance compared to PCOS women without hyperandrogenism [60].

Regarding the findings of the AHEI-2010, this is the only one of the five indices which establishes the maximum score for whole grains at 75 g/day. This quantity much reduces glycemic load compared to the other indices. The other different item is sugar-sweetened beverages and fruit juice. AHEI-2010 scored 0 points if people drank one or more servings per day and the maximum score when subjects did not drink any sugar-sweetened beverages and fruit juice. The DASH score has an item for sweetened beverages as well, but it is based on quintiles intake. A subject drinking sweetened beverages would be more penalized in the AHEI-2010 than in the DASH index. We also explored if a higher consumption of any single food group of AHEI-2010 index could explain the inverse association. Only vegetables intake was close to statistical significance. Thus, the effect of the AHEI-2010 dietary pattern in the PCOS phenotype with hyperandrogenism and oligovulation may have a synergic effect of food groups and not be an isolated action of just one group. Moreover, compared to individual food items, dietary patterns measured a priori via dietary indices can incorporate complex interactions among foods/nutrients and can better reduce the risk of some diseases [61].

In our case, we did not find a clear explanation for the inverse relationship between DASH index and the O + POM phenotype, but the relatively low number of subjects in this phenotype may play a role in it. Therefore, we cannot rule out a chance finding may occur.

Unexpectedly, the highest quartile of DASH index was positively associated with the O + POM phenotype, increasing the risk of having this phenotype with a higher DASH adherence. Contrarily, the DASH diet may improve BMI and other metabolic parameters which could reduce PCOS symptomatology. Three studies have shown that adherence to the DASH diet between eight and 12 weeks among PCOS women could have beneficial effects on BMI, androstenedione, anti-Müllerian hormone (AMH), insulin and lipid metabolism [31,32]. An interventional study with the DASH diet for eight weeks led to a significant reduction in serum insulin, triglycerides and very-low-density lipoprotein cholesterol, and a significant increase in total antioxidant capacity, glutathione levels [34] and c-reactive protein [33]. The DASH diet is characterized by consumption of seafood, poultry, whole grains, fruits, and vegetables, which have been related to better fertility in women and better semen quality in men [62].

Another point to consider is the importance of fibre on PCOS because of its capacity in acting on gut microbiota as a prebiotic. An increment of gut permeability, the reduction of biodiversity and a growing endotoxemia by lipopolysaccharide of gram negative bacteria promote higher absorption of energy, activate the immune system and inflammation, and increase hyperandrogenism and insulin resistance [63]. The gut microbiota has been implicated to play a critical role in metabolic diseases such as PCOS, and may modulate the secretion of mediators of the brain–gut axis [64]. However, the research is still in its early stage [65,66,67,68]. Fibre and microbiome may be a confounding factor in the present study. However, fibre did not accomplish statistical criteria for inclusion in our analyses, whereas carbohydrates did. Fibre and carbohydrates intakes presented a high colinearity, for including both.

On the other hand, our study subjects were incident and prevalent PCOS cases. Therefore, some women could have recently changed their dietary patterns due to their symptoms. In fact, healthy dietary pattern adherence and physical exercise are the first line of treatment recommended for PCOS patients [69,70,71,72,73]. However, the monitoring of these changes in diet and physical activity is not integrated in the usual clinical practice [69,74]. Moreover, there are few studies that have explored diet changes after PCOS diagnosis, and their results are inconclusive [58,75,76,77,78].

Lastly, we have not found studies evaluating associations of diet among PCOS phenotypes. In fact, clinical manifestations of PCOS are notably different among PCOS phenotypes, such as those presenting androgenism or not, although all of them take part of the same syndrome [79].

To our knowledge, our study is the first one analyzing dietary indices among different PCOS phenotypes according to the Rotterdam criteria, instead of other classifications such as anovulatory and ovulatory PCOS. Our modest findings may reveal that it would be necessary to make dietary recommendations based on PCOS phenotypes according to physiological differences. Likewise, it has been reported that a phenotypic approach would be highly convenient for clinical practice and epidemiologic research [73,80].

Furthermore, the relation between diet and fertility in women has been studied in the last three decades [81]. Based on the Nurse Health Study cohort, a Fertility Diet was described, showing that its adherence was able to reduce the attributable risk of ovulatory disorder infertility in 66%of cases (95% CI: 29–86%) [82]. Relationships between AHEI and DASH indices and PCOS have been evaluated, but mostly in randomized controlled trials seeking to improve PCOS symptoms, mainly using low carbohydrate high proteins diets [83,84]. In women with PCOS, it would be worthwhile to carry out interventional studies with a mix approximation: firstly, to follow a high protein low carbohydrate diet, and secondly, a Mediterranean Diet. A high protein, low carbohydrate, diet may allow a quick reduction of body weight, hyperinsulinemia and inflammation. The Mediterranean Diet may be the second step for maintenance of the previous results with less secondary effects, and an increment of adherence to a long-term healthier diet. For example, the PREDIMED study was one of the most important nutritional interventions carried out in Spain. PREDIMED showed a 30% CVD risk reduction with the Mediterranean diet, which is of similar magnitude to that reported in statin trials [85].

The current study has some limitations. First, we used a FFQ as dietary assessment method that has limitations but was comparable to other tools for assessment of food intake in nutritional epidemiology studies. In addition, all self-reported dietary intakes are subject to misreporting, with different types of foods likely to be either over or under-reported [86]. However, we used a validated FFQ with a mean correlation coefficient for nutrient intakes equal to 0.40 for reproducibility and 0.47 for validity. Second, we applied dietary indices which were not specifically created for the Spanish population, except the rMED index that was implemented for the Spanish cohort of European Prospective Investigation into Cancer (EPIC). This may limit external validity of this specific tool. However, all these dietary indices have already been used in multiple nutritional investigations worldwide. Third, case-control studies present a higher risk of selection and information bias than other study designs. Nevertheless, we selected both cases and controls from the same population of women who visited the same medical services in the same time period: a public hospital and outpatient clinics. Concerning information bias, if misclassification of PCOS status may have occurred, it would have contributed to underestimating the true magnitude of the relationship. Finally, we employed the Rotterdam criteria, which are widely used in gynecological studies and clinical practice.

In conclusion, we did not find any clear association between dietary indices (AHEI, AHEI 2010, AMED, RMED and DASH) and PCOS and its phenotypes. We only observed an inverse association between AHEI -2010 score and less probabilities of the H + O phenotype, but this finding must be taken cautiously. To our knowledge, our study is the first evaluating the association between dietary indices and four different phenotypes of PCOS, according to the Rotterdam criteria, and using Mediterranean Diet indices (aMED and rMED) to assess dietary patterns in women with PCOS. Further adequately powered studies are required to clarify if there are associations between dietary indices and PCOS phenotypes. In addition, prospective studies to monitor and evaluate the effect of changes in diet in PCOS women during regular clinical practice are necessary, which is one of the first treatment options for many women with PCOS according to the international practice guidelines.

## Figures and Tables

**Figure 1 nutrients-13-00313-f001:**
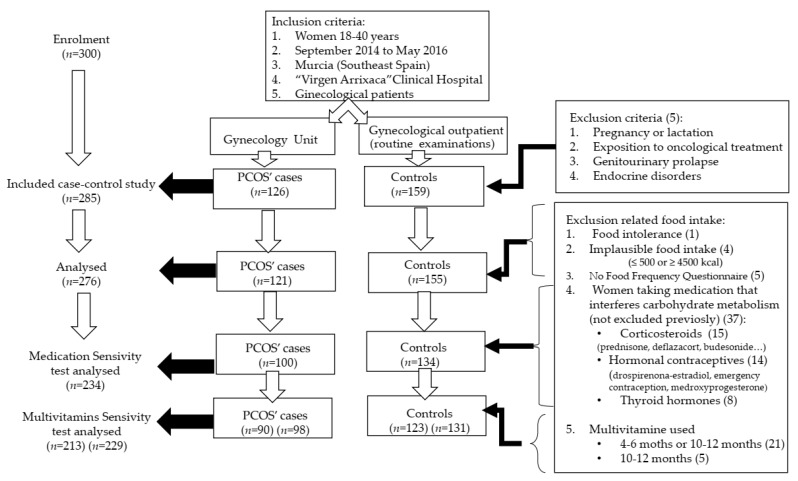
Flow chart of inclusion and exclusion criteria of polycystic ovary syndrome (PCOS) case-control study of southeast of Spain from 2014 to 2016.

**Table 1 nutrients-13-00313-t001:** Description in detail of dietary indices.

Category of Scoring Dietary Indices	AHEI ^1^		AHEI-2010		aMED		rMED		DASH	
	Min ^2^ (0)	Max ^3^(10)	Min (0)	Max (10)	Min (0)	Max (1)	Min (0)	Max (2)	Min (0)	Max (5)
Vegetables	0	5 servings/day	0	≥5 servings/day	<median servings/day	>median servings/day	Lowest tertile	Greatest tertile	Lowest quintile	Greatest quintile
Fruit	0	4 servings/day	0	>4 servings/day	<median servings/day	>median servings/day	Lowest tertile	Greatest tertile	Lowest quintile	Greatest quintile
Nuts	0	1 servings/day			<median servings/day	>median servings/day	Lowest tertile	Greatest tertile		
Nuts and legumes			0	≥1 servings/day					Lowest quintile	Greatest quintile
Legumes					<median servings/day	>median servings/day	Lowest tertile	Greatest tertile		
Cereals										
Cereal fiber	0	15 g/day								
Whole grains			0	75 g/day	<median servings/day	>median servings/day			Lowest quintile	Greatest quintile
Cereals							Lowest tertile	Greatest tertile		
Meat										
Ratio of white to red meat	0	4								
Red and processed meat			≥1.5	0 servings/day	>median servings/day	<median servings/day				
Meat							Greatest tertile	Lowest tertile		
Fats										
Trans fat	≥4	≤0.5% of energy	≥4	≤0.5% of energy						
Ratio of saturated to polyunsaturated	≥4	≥1								
Ratio of monosaturated to saturated					<median servings/day	>median servings/day				
Polyunsaturated			≤2	≥10% of energy						
ω 3 fats (EPA + DHA) ^4^			0	250 mg/day						
Olive oil							Lowest tertile	Greatest tertile		
Fish					<median servings/day	>median servings/day	Lowest tertile	Greatest tertile		
Dairy products							Greatest tertile	Lowest tertile		
Low-fat dairy									Lowest quintile	Greatest quintile
Sodium			Highest decile	Lowest decile (mg/day)					Greatest quintile	Lowest quintile
Alcohol	0 or >2.5 servings/day	0.5–1.5 servings/day		0.5–1.5 drinks/day	<5 or <25 d/day	5–25 d/day				
Sugar-sweetened beverages and fruit juices			≥1	0						
Total maximum score		87.5		110		9		18		40

^1^ AHEI-2010, Alternate Healthy Index 2010; AHEI, Alternate Healthy Index; aMED, Alternate Mediterranean Diet Score; rMED, Relative Mediterranean Diet Score; DASH, Dietary Approaches to Stop Hypertension.^2^ Criteria for minimum score, ^3^ Criteria for maximum score. ^4^ EPA, Eicosapentaenoic acid; DHA, Docosahexaenoic acid.

**Table 2 nutrients-13-00313-t002:** Demographic characteristics and nutrient intake according to quintiles of adherence to dietary quality indices (*n* = 275).

Median Value	AHEI			AHEI-2010			aMED			rMED			DASH		
	Q1 (13–32) *n* = 70	Q4 (48–78) *n* = 68	*p*-value ^1^	Q1 (27–56) *n* = 69	Q4 (72–97) *n* = 68	*p*-value	Q1 (0–3) *n* = 69	Q4 (7–10) *n* = 55	*p*-value	Q1 (2–9) *n* = 165	Q4 (13–15) *n* = 27	*p*-value	Q1 (11–19) *n* = 74	Q4 (28–35) *n* = 61	*p*-value ^1^
Age (years)	27.5 (23.0; 32.0)	31.0 (26.0; 34.8)	0.07	28.0 (23.5; 32.0)	33.0 (29.0; 35.0)	0.00	27.0 (23.0; 32.0)	30.0 (26.0; 34.0)	0.01	29.0 (24.0;33.0)	31.0 (26.0;34.0)	0.43	31.0 (23.8; 35.0)	28.0 (24.0; 33.0)	0.25
BMI (kg/m^2^)	25.2 (21.4; 32.3)	21.6 (20.1; 23.9)	0.00	25.9 (20.5; 30.9)	21.7 (20.1; 23.7)	0.01	24.2 (20.9; 30.9)	21.9 (20.6; 24.6)	0.00	22.7 (20.1;27.5)	22.0 (19.7;24.3)	0.14	23.2 (20.8; 28.1)	21.9 (20.4; 26.6)	0.91
Calories intake (Kcal)	1537.5 (1230.6; 1972.7)	2188.3 (1723.1; 2847.6)	0.00	1908.1 (1448.5; 2310.5)	1737.8 (1420.7; 2358.6)	0.67	1456.9 (1178.5; 1937.7)	2131.7 (1672.4; 2928.4)	0.00	1945.3 (1605.1;2384.6)	1466.2 (1288.2;1832.3)	0.00	1338.0 (1086.8; 1570.6)	2775.2 (2211.5; 3307.3)	0.00
Physical activity (h/week)	3.6 (0.5; 11.9)	8.4 (5.5; 14.0)	0.00	5.0 (2.0; 14.0)	8.0 (5.3; 14.3)	0.00	4.7 (0.6; 10.8)	8.0 (5.5; 13.8)	0.00	6.0 (2.3;13.7)	4.3 (0.0; 8.5)	0.01	5.0 (0.8; 10.3)	8.0 (3.1; 15.9)	0.17
Alcohol (g/day)	0.4 (0.0; 1.4)	3.6 (1.9; 6.3)	0.00	0.5 (0.0; 1.7)	3.6 (1.3; 6.6)	0.00	0.9 (0.0; 2.4)	5.8 (2.9; 8.1)	0.00	1.3 (0.0;3.5)	7.0 (3.8; 10.0)	0.00	1.3 (0.5; 5.9)	2.9 (0.6; 5.4)	0.64
Caffeine (mg/day)	31.2 (8.3; 49.4)	48.3 (18.5; 77.1)	0.02	33.4 (13.5; 55.2)	51.7 (22.4; 81.3)	0.04	35.4 (13.2; 56.2)	56.6 (22.0; 94.8)	0.05	37.0 (14.7;68.4)	46.3 (20.4; 99.0)	0.21	37.3 (12.5; 67.3)	47.3 (18.0; 79.9)	0.46
Carbohydrate (g/day)	158.5 (131.4; 181.5)	192.4 (170.9; 207.2)	0.00	170.5 (137.3; 196.1)	190.5 (167.8; 207.0)	0.00	165.0 (132.8; 193.6)	186.8 (164.5; 198.2)	0.01	175.0 (149.7;200.4)	167.2 (155.4; 193.5)	0.83	163.4 (139.3; 189.1)	185.3 (166.7; 204.0)	0.00
Saturated fats (g/day)	22.1 (19.7; 25.3)	17.9 (14.2; 21.4)	0.00	22.5 (19.9; 26.0)	16.6 (13.8; 19.7)	0.00	21.3 (19.2; 24.9)	18.8 (16.3; 22.1)	0.00	21.2 (18.5;24.4)	17.9 (15.6; 21.4)	0.00	19.7 (16.2; 22.5)	21.6 (19.3; 25.4)	0.02
Omega-3 (mg/day)	1.3 (1.2; 1.4)	1.7 (1.5; 2.2)	0.00	1.3 (1.1; 1.5)	1.6 (1.4; 2.0)	0.00	1.2 (1.0; 1.4)	1.7 (1.5; 2.2)	0.00	1.5 (1.3;2.0)	1.4 (1.1; 1.5)	0.00	1.3 (1.1; 1.5)	1.8 (1.5; 2.2)	0.00

^1^ Kruskal-Wallis tests were used to test for associations between the level of diet indeces.

**Table 3 nutrients-13-00313-t003:** Multivariate adjusted 1 Odds Ratios (ORs) between dietary indices and total, anovulatory and ovulatory PCOS (total *n* = 276; cases = 121, controls n = 155).

Range for EachQuartile of Index ^2^		Total PCOS			Anovulatory			Ovulatory		
		Cases = 121			Cases = 88			Cases = 33		
		OR ^1^	95%CI	*p*-Value	OR	95%CI	*p*-Value	OR	95%CI	*p*-Value
AHEI2010										
Q1 (27–56)		Ref.			Ref.			Ref.		
Q2 (57–63)		1.0	(0.5; 2.0)	0.93	1.1	(0.5; 2.3)	0.81	0.8	(0.3; 2.3)	0.69
Q3 (64–71)		0.6	(0.3; 1.2)	0.14	0.5	(0.2; 1.1)	0.09	1.1	(0.4; 2.9)	0.89
Q4 (72–97)		0.7	(0.3; 1.6)	0.44	0.9	(0.4; 2.0)	0.79	0.7	(0.2; 2.1)	0.50
	*p*-value _for trend_		0.41			0.24			0.84	
AHEI										
Q1 (13–32)		Ref.			Ref.			Ref.		
Q2 (33–40)		1.6	(0.8; 3.5)	0.22	1.8	(0.8; 3.9)	0.17	1.0	(0.4; 2.8)	1.00
Q3 (41–47)		1.0	(0.5; 2.3)	0.94	1.2	(0.5; 2.8)	0.66	0.8	(0.2; 2.4)	0.65
Q4 (48–78)		0.8	(0.3; 2.0)	0.65	1.0	(0.4; 2.6)	0.97	0.7	(0.2; 2.2)	0.50
	*p*-value _for trend_		0.31			0.41			0.86	
aMED										
Q1 (0–3)		Ref.			Ref.			Ref.		
Q2 (4)		0.6	(0.3; 1.2)	0.16	0.8	(0.4; 1.7)	0.58	0.5	(0.2; 1.5)	0.23
Q3 (5–6)		0.9	(0.4; 2.0)	0.86	1.2	(0.5; 2.6)	0.70	0.7	(0.2; 2.0)	0.50
Q4 (7–10)		0.8	(0.4; 2.0)	0.68	0.8	(0.3; 2.1)	0.71	1.0	(0.3; 3.3)	0.97
	*p*-value _for trend_		0.48			0.75			0.55	
rMED										
Q1 (2–9)		Ref.			Ref.			Ref.		
Q2 (10)		0.9	(0.4; 2.0)	0.82	1.1	(0.5; 2.5)	0.82	0.7	(0.2; 2.3)	0.59
Q3 (11–12)		0.6	(0.3; 1.3)	0.19	0.9	(0.4; 2.0)	0.74	0.3	(0.1; 1.5)	0.15
Q4 (13–15)		1.6	(0.7; 3.9)	0.31	1.6	(0.6; 4.1)	0.33	1.1	(0.3; 3.8)	0.85
	*p*-value _for trend_		0.33			0.73			0.48	
DASH										
Q1 (11–19)		Ref.			Ref.			Ref.		
Q2 (20–23)		0.8	(0.4; 1.7)	0.58	1.0	(0.5; 2.3)	0.98	0.6	(0.2; 1.9)	0.44
Q3 (24–27)		1.1	(0.5; 2.5)	0.84	0.9	(0.4; 2.3)	0.90	1.3	(0.4; 3.9)	0.66
Q4 (28–35)		1.6	(0.5; 4.7)	0.39	2.9	(0.9; 9.3)	0.07	0.4	(0.1; 2.0)	0.26
	*p*-value _for trend_		0.51			0.09			0.26	

^1,2^ Adjusted for calories intake (kcal/day), BMI, moderate-vigorous physical activity (h/week), caffeine intake (mg/day) and carbohydrates intake (g/day). AHEI-2010, Alternate Healthy Index 2010; AHEI, Alternate Healthy Index; aMED, Alternate Mediterranean Diet Score; rMED, Relative Mediterranean Diet Score; DASH, Dietary Approaches to Stop Hypertension.

**Table 4 nutrients-13-00313-t004:** Multivariate adjusted 1 ORs between dietary indices and phenotypes 3−5 of PCOS (*n* = 276).

Index	H + O + POM ^2^			H + O ^3^				H + POM ^4^			O + POM ^5^		
	Cases = 52			Cases = 18				Cases = 33			Cases = 18		
	OR ^1^	95%CI	*p*-Value	OR		95%CI	*p*-Value	OR	95%CI	*p*-Value	OR	95%CI	*p*-Value
AHEI-2010													
Q1 (27–56)	Ref.			Ref.				Ref.			Ref.		
Q2 (57–63)	1.5	(0.7; 3.5)	0.34	0.2		(0.0; 0.7)	0.01	0.8	(0.3; 2.3)	0.69	4.9	(0.9; 25.6)	0.06
Q3 (64–71)	0.7	(0.3; 1.8)	0.43	0.1		(0.0; 0.9)	0.04	1.1	(0.4; 2.9)	0.89	2.5	(0.4; 16.7)	0.35
Q4 (72–97)	1.1	(0.4; 2.9)	0.85	0.2		(0.0; 1.2)	0.08	0.7	(0.2; 2.1)	0.50	4.9	(0.8; 30.4)	0.09
*p*-value _for trend_		0.42				0.02			0.84			0.24	
AHEI													
Q1 (13–32)	Ref.			Ref.				Ref.			Ref.		
Q2 (33–40)	2.3	(0.9; 6.3)	0.09	0.4		(0.1; 1.5)	0.17	1.0	(0.4; 2.8)	1.00	2.9	(0.7; 12.9)	0.15
Q3 (41–47)	2.7	(1.0; 7.3)	0.06	0.0		(0.0; −)	1.00	0.8	(0.2; 2.4)	0.65	1.8	(0.4; 8.8)	0.48
Q4 (48–78)	1.7	(0.5; 5.5)	0.38	0.3		(0.0; 1.6)	0.15	0.7	(0.2; 2.2)	0.50	1.5	(0.2; 9.5)	0.69
*p*-value _for trend_		0.21				0.41			0.86			0.48	
aMED													
Q1 (0–3)	Ref.			Ref.				Ref.			Ref.		
Q2 (4)	1.5	(0.6; 3.7)	0.38	0.2		(0.0; 0.9)	0.04	0.5	(0.2; 1.5)	0.23	1.1	(0.2; 5.8)	0.90
Q3 (5–6)	1.4	(0.6; 3.7)	0.46	0.3		(0.1; 1.4)	0.13	0.7	(0.2; 2.0)	0.50	3.6	(0.8; 16.8)	0.11
Q4 (7–10)	1.1	(0.4; 3.5)	0.86	0.2		(0.0; 1.8)	0.15	1.0	(0.3; 3.3)	0.97	3.2	(0.5; 18.7)	0.20
*p*-value _for trend_		0.78				0.12			0.55			0.28	
rMED													
Q1 (2–9)	Ref.			Ref.				Ref.			Ref.		
Q2 (10)	1.1	(0.4; 2.9)	0.86	0.7		(0.1; 6.3)	0.77	0.7	(0.2;2.3)	0.59	1.4	(0.4; 5.6)	0.63
Q3 (11–12)	0.8	(0.3; 2.1)	0.60	1.5		(0.4; 5.6)	0.57	0.3	(0.1; 1.5)	0.15	0.5	(0.1; 2.4)	0.35
Q4 (13–15)	1.9	(0.6; 5.4)	0.25	1.6		(0.3; 9.1)	0.61	1.1	(0.3; 3.8)	0.85	0.6	(0.1; 4.9)	0.60
*p*-value _for trend_		0.60				0.88			0.48			0.68	
**Index**	**H + O + POM**			**H + O**			**H + POM**			**O + POM**			
	**Cases = 52**			**Cases = 18**			**Cases = 33**			**Cases = 18**			
	**OR**	**95%CI**	***p*-Value**	**OR**	**95%CI**	***p*-Value**	**OR**	**95%CI**	***p*-Value**	**OR**	**95%CI**		***p*-Value**
DASH													
Q1 (11–19)	Ref.			Ref.			Ref.			Ref.			
Q2 (20–23)	1.0	(0.4; 2.6)	0.96	1.1	(0.3; 4.3)	0.84	0.6	(0.2; 1.9)	0.44	0.8	(0.1; 5.1)		0.79
Q3 (24–27)	0.8	(0.2; 2.4)	0.64	0.3	(0.1; 2.3)	0.27	1.3	(0.4; 3.9)	0.66	4.6	(0.9; 24.4)		0.08
Q4 (28–35)	3.2	(0.9; 11.9)	0.08	0.0	(0.0; −)	1.00	0.4	(0.1; 2.0)	0.26	9.2	(1.1; 74.7)		0.04
*p*-value _for trend_		0.07			0.84			0.26			0.05		

^1^ Adjusted for calories intake (kcal/day), BMI, moderate-vigorous physical activity (h/week), caffeine intake (mg/day) and carbohydrates intake (g/day). AHEI-2010, Alternate Healthy Index 2010; AHEI, Alternate Healthy Index; aMED, Alternate Mediterranean Diet Score; rMED, Relative Mediterranean Diet Score; DASH, Dietary Approaches to Stop Hypertension. ^2^ H + O + POM “Hyperandorgenism + Oligo/amenorrhea + Polycystic ovaries morphology” phenotype. ^3^ H + O “Hyperandorgenism + Oligo/amenorrhea” phenotype. ^4^ H + POM “Hyperandorgenism + Polycystic Ovary Morphology” phenotype. ^5^ O + POM “Oligo/amenorrhea + Polycystic Ovary Morphology” phenotype.

## Data Availability

The data that support the findings of this study are restricted for research use only. The data are not publicly available. Data are available from the authors upon reasonable request and with permission from the Departments of Preventive Medicine and Obstetrics and Gynecology, University Clinical Hospital Virgen de la Arrixaca, Spain.

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
