# Peer review of "Are Dietary Indices Associated with Polycystic Ovary Syndrome and Its Phenotypes? A Preliminary Study"

_nutrients, 2021, doi:10.3390/nu13020313_

Round 1
Reviewer 1 Report
Revision of manuscript: " Are dietary indices associated with polycystic ovary syndrome and its phenotypes? A case-control study”
This is an interesting case-control study on the influence associations of dietary indices and polycystic ovary syndrome and its phenotypes. One of the benefits and advantages of this study is that they take into account significantion of healthy dietary patterns and physical exercise as the first line of recommended treatment for PCOS patients
The methods of the study are well-described, and most of the inclusion and exclusion criteria clearly stated. Also, the statistical analyses are precisely explained. The topic is important and most of the issues are discussed and explained in the discussion section.
However, there are some serious limitations in the manuscript which should be corrected.
First of all in exclusion criteria section authors should also added informations about digestion diseases and food allergy (of course this criterias should eliminated the participants from the study). What is more, we should remember that most of the PCOS have laso insulin resistance or even type 2 diabates. I think that authors should recruit only women who don’t taking any mediacation to regulation carbohydrate metabolism (we don’t have this information in Materials and Methods).
Information about amount of participants are unclear. Please, check the cases in the table.
The title of the article should be change. Beacuse of not many participants in soubgroups, the better option of the title is: „Are dietary indices associated with polycystic ovary syndrome and its phenotypes? A preliminary study”
Author Response
Answers to Reviewers Comments
The responses to the individual comments of four reviewers are detailed below.
(Note: Reviewer comments are in italic, authors' responses are in normal text).
The authors would like to thank all of the reviewers for precise and thoughtful
comments and constructive criticism which has led to a better manuscript. Below we
respond to each referee comments individually. Reviewers please note that figure, table
and page numbers refer to figures, table and pages in the corrected manuscript
Comments from Reviewer 1
We value the comments received greatly, as they have pointed out a number of issues to
be addressed.
Comment 1:
“First of all in exclusion criteria section authors should also added informations about digestion diseases and food allergy (of course this criterias should eliminated the participants from the study). What is more, we should remember that most of the PCOS have aSso insulin resistance or even type 2 diabates. I think that authors should recruit only women who don’t taking any medication to regulation carbohydrate metabolism (we don’t have this information in Materials and Methods)”
Response 1: We agree with this and have incorporated your suggestion in Materials and Methods and we have repeated all analyses excluding, subjects (sensitivity analysis) with digestive diseases (0), food allergy (1), insulin resistance (2, without medication), type 2 diabetes, autoimmune disease or women who don’t taking any medication to alters carbohydrate metabolism (32) (10% of prevalent of side effect related to carbohydrate metabolism) (additionally to previous works and our first tables of result). The tables above indicate exclusion criteria and the number of subjects for each disease or medication
Table R1. Exclusion criteria and the number of subjects for diseases or medications that alters carbohydrate metabolism
|
Exclusion criteria |
n |
% |
|
Food Intolerance |
1 |
0.3 |
|
Insuline resistente |
2 |
0.7 |
|
Contraceptives |
4 |
1.3 |
|
Corticosteroids |
16 |
5.3 |
|
Thyroid hormones |
12 |
4.0 |
|
TOTAL |
32 |
6.3 |
Table R2. Specific medicament which alters carbohydrate metabolism excluded from the sample and the number of subjects for each category
|
Group |
Medicament |
n |
% |
|
Contraceptives |
Emergency contraception |
2 |
0,7 |
|
Contraceptives |
ethinyl estradiol-drospirenone Combined Oral Contraceptive |
1 |
0.3 |
|
Contraceptives |
Medroxyprogesterone acetate |
1 |
0.3 |
|
Corticosteroids |
Betamethasone and gentamicin |
2 |
0.7 |
|
Corticosteroids |
Budesonide |
2 |
0.7 |
|
Corticosteroids |
Deflazacort |
3 |
1.0 |
|
Corticosteroids |
Prednisone |
3 |
1.0 |
|
Corticosteroids |
Methylprednisolone |
4 |
1.3 |
|
Corticosteroids |
Methylprednisolone |
1 |
0.3 |
|
Corticosteroids |
Deflazacort |
1 |
0.3 |
|
Thyroid hormones |
Levothyroxine |
12 |
4.0 |
|
|
TOTAL |
32 |
6.0% |
|
|
Total sample recruitmen |
300 |
100% |
Comment 2:
Information about amount of participants are unclear. Please, check the cases in the table.
Response 2: Thank you for pointing that out. All numbers have been reviewed and changed accordingly in order to be consistent throughout the manuscript and we have added a flow chart.
Comment 3: The title of the article should be change. Beacuse of not many participants in soubgroups, the better option of the title is: „Are dietary indices associated with polycystic ovary syndrome and its phenotypes? A preliminary study”
Response 3: Agree. We have modified the title accordingly (line 3)

Reviewer 2 Report
Thank you for the opportunity to read the manuscript, exploring associations of dietary patterns measured via dietary indices and PCOS with respect to its phenotypes. The obtained results in quite a big study sample demonstrate only few significant associations/findings; however, the study design is convincing, and the study is generally well described in all the necessary details. The topic is correctly introduced, the results are clearly presented (tables could be more commented…) and then appropriately discussed. Especially the discussion is well written and will help readers, not specified in this field, to understand the results.
I have no serious concern to mention. There are plenty of minor mistakes in spelling, abbreviations (not explaining them when first used), and English. Some sentences are a bit hard to understand or could be restructured (for example line 118 – placing the frequency options into brackets would help). Line 165 is not correct – “differences…were different”. In line 172: was it really significant if p=0.081?
Author Response
Answers to Reviewers Comments
The responses to the individual comments of four reviewers are detailed below.
(Note: Reviewer comments are in italic, authors' responses are in normal text).
The authors would like to thank all of the reviewers for precise and thoughtful
comments and constructive criticism which has led to a better manuscript. Below we
respond to each referee comments individually. Reviewers please note that figure, table
and page numbers refer to figures, table and pages in the corrected manuscript
Comments from Reviewer 2
We highly appreciate the reviewer comments
Comment 2-1: “… presented (tables could be more commented…) and then appropriately discussed”.
Response 2-1: We have commented more the results in previous versions (see below) but we would like to know if this way may be better or not.
Contestar después de repetir los análisis quitando mujeres con enfermdedes, medicaciones, y … ¿multivitaminas?
Ver si vale algo de lo siguiente que descartamos para hacerlo más breve
In the multivariable analysis, we did not find any associations between the diet scores and PCOS or ovulatory and anovulatory PCOS types. Only, a borderline positive association between DASH index and anovulatory PCOS type (P for trend =0.07) due an increment of the probabilities of PCOS between the highest quartile respect to the lowest one (ORQ4vsQ1 = 2.9; 95%CI : 0.9–9.3; p =0.07)
In the analyses by PCOS subtypes, women with higher adherences to the AHEI-2010 pattern were less likely to present PCOS, specifically “Hyperandrogenism + Oligo/amenorrhea” subtype (P for trend= 0.02) (Table 3). Women in the lower-intermediate (Q2) and upper-intermediate quartiles (Q3) of AHEI-2010 score have 80% (OR = 0.2; 95%CI : 0.0–0.7; p=0.01) and 90% (OR = 0.1; 95%CI : 0.0–0.9; p=0.02) less probabilities of PCOS with “Hyperandrogenism + Oligo/amenorrhea”, than the lowest category (Q1), respectively. This association slightly decrease in the highest quartile (Q4 vs Q1; OR = 0.2; 95%CI : 0.0–1.2; p=0.08). Similarly, subjects in the second quintile of amed present less probabilities of the same PCOS subtype (OR = 0.2; 95%CI : 0.0–0.9; p=0.04).
Furthermore, we observed an inverse lineal trend between DASH index and “oligo/amenorrhea+ polycystic ovary morphology “subtype (ORQ4vsQ1 = 9.2; 95%CI : 1.1–74.7; P for trend =0.05). In parallel, the same relation was borderline between DASH index and the same PCOS subtype plus hyperandrogenism (“Hyperandrogenism + oligo/amenorrhea+ polycystic ovary morphology ”) (P for trend =0.07).
Comment 2-2: “There are plenty of minor mistakes in spelling…..and English”
Response 2-2: All spelling and grammatical errors pointed out by the reviewer have been corrected.
Comment 2-3: “….mistakes in spelling abbreviations (not explaining them when first used), …”
Response 2-3: We have explained abbreviation when first used in:
Line 34-35 Hyperandrogenism + Oligoanovulation instead of “H+O”
Line 61 alternate Mediterranean Diet Score (aMED), relative Mediterranean Diet Score (rMED)
Line 62 Alternate Healthy Index (AHEI) and Dietary Approaches to Stop Hypertension (DASH)
Line 128 United States instead of US
Line 171 Body Mass Index (BMI)
Line 204 Odds Ratios (ORs)
Comment 2-4: “Some sentences are a bit hard to understand or could be restructured (for example line 118 – placing the frequency options into brackets would help)”
Response 2-4: We have changed the text following your suggestion:
“Subjects had to choose one of the nine options about how often, on average, they had consumed each food ítem (from never or less than once a month, to six or more times a day)”
Comment 2-5: Line 165 is not correct – “differences…were different”.
Response 2-5: “Differences between median values scores across quartiles of rMED index score were different compared to the other dietary indices” have corrected by “Median values scores across quartiles of rMED index score were different compared to …”
Comment 2-6: In line 172: was it really significant if p=0.081?
Response 2-6: This association was only conducted by vegetables food item of AHEI-2010 index, but not significantly (P for trend= 0.081).

Reviewer 3 Report
The manuscript is well-written and the authors have performed an interesting study on the association between dietary indices and total, anovulatory and ovulatory PCOS. The study was done on 276 women from Spain and the food frequency questionnaire were considered for analysis. The did not find statistical significance between diet and PCOS.
- It will be interesting to see if there were any common food taken by the individuals in different diet based on the reports in the food frequency questionnaire.
- Did the women take dietary supplements like multivitamins or did the authors observe any common variable in different diet that can be used as a feature to be corrected in the statistical analysis?
- Are there any known dietary intervention studies that has been used for observing the effect of diet in the PCOS individuals?
- Minor correction: line 252: "high range of confidence" instead of "high rage of confidence".
Author Response
Answers to Reviewers Comments
The responses to the individual comments of four reviewers are detailed below.
(Note: Reviewer comments are in italic, authors' responses are in normal text).
The authors would like to thank all of the reviewers for precise and thoughtful
comments and constructive criticism which has led to a better manuscript. Below we
respond to each referee comments individually. Reviewers please note that figure, table
and page numbers refer to figures, table and pages in the corrected manuscript
Comments from Reviewer 3
The authors would like to thank deeply the reviewer for his time and effort.
Comment 3-1: “It will be interesting to see if there were any common food taken by the individuals in different diet based on the reports in the food frequency questionnaire”
Response 3-1: Yes, we agree. We would like to have a bit more time to do these new analyses, this gave us a better understanding of the relation between food intake and PCOS. However, our experience it easier find statistical significative association using food groups such as the components of dietary indices than single foods.
Comment 3-2: Did the women take dietary supplements like multivitamins or did the authors observe any common variable in different diet that can be used as a feature to be corrected in the statistical analysis?
Response 3-2: Yes, the use of multivitamin should be taken into account in the manuscript as comment reviewer 3 because the prevalence is from 2 % to 10.9% depends of time period taking supplement. Firstly, we did not consider because dietary supplements use is much less common in Spain than in other countries such as United States. The next table summarizes prevalence of using multivitamins and other supplements such as probiotics, minerals, brewer’s yeast during “4-6 or 10-12 months” or only “10-12 months”
Prevalence of using multivitamins and other supplements such as probiotics, minerals, brewer’s yeast during “4-6 or 10-12 months” or only “10-12 months”
|
Time categories |
n |
% |
|
4-6 or 10-12 months |
30 |
10,9 |
|
10-12 months |
7 |
2,5 |
|
Sample size |
276 |
100,0 |
|
4-6 or 10-12 months |
21 |
9,0 |
|
10-12 months |
5 |
2,1 |
|
Sample size |
234 |
100,0 |
We have carried out the follow sensivity tests excluding women taking multivitamins and other dietary supplements:
During 4-6 or 10-12 months with a sample size of 276 (presented initially in the papter)
During only 10-12 months with a sample size of 276
During 4-6 or 10-12 months with a sample size of 234 (excluding women who takes medication that alters carbohydrate metabolism, according reviewer 1’s comments)
During only 10-12 months with a sample size of 234
Figures 1 may help understanding these option and changes.
|
Figure 1. Flow chart of exclusion criteria and sensivity tests excluding use of multivitamins and other dietary supplements |
.
There were no changes in the association p-values when we consider multivitamins as covariable in statistical analyses with a sample size of 276 or 234
If the reviewers consider we can change all the figures in Tables and results for incorporating “multivitamins” as covariable in statistical analyses
Comment 3-3: “Are there any known dietary intervention studies that has been used for observing the effect of diet in the PCOS individuals?”
Response 3-3: Yes, most of dietary intervention studies are focus in high protein low carbohydrates diets, describe in lines 61-69 and references: 25-28, 30, 77 and 78. There are 3 interventions with DASH diet (lines 69-72, 244- 251; references 31-34)
Comment 3-4: “Minor correction: line 252: "high range of confidence" instead of "high rage of confidence".
Response 3-4: It has been changed

Reviewer 4 Report
In the current study the potential association of 5 dietary indices with Polycystic ovary syndrome was questioned as well as their predictive ability to its different phenotypes. Prior dietary habits were assessed utilizing a semi-quantitative food frequency questionnaire. Resulting indices were categorized in quartiles using the lowest as reference, and following adjustments for confounders, no association was found with any PCOS diagnostic criteria. However, a relationship was noted with the anovulatory PCOS phenotypes, particularly “H+O” phenotype positively related to the AHEI-2010 diet and an inverse correlation based on the DASH diet score with the “O+POM” PCOS’ phenotype.
Interesting and of potential clinical importance, as the relationship between dietary patterns and metabolic health in specific PCOS phenotypes may be highly relevant for personalized treatment approaches. However, there are significant concerns about the study design that weaken the conclusions and implications of this paper. The first main issue of this manuscript is the sample size that was generally small, and further divided into multiple subgroups. While one rule of thumb may be to take a sample of 10 times more than the no. of questions for 2 major study groups, I suggest the authors to report the calculations of sample size in the method section which allowed for further subgrouping while maintaining a sufficiently high study power, based also on prevalence studies of predominant PCOS phenotypes among the overall population;
- The second main issue is the lack of an in depth description of the hormonal and inflammatory values of the study groups. As mentioned in the introduction, PCOS is a lifelong reproductive and metabolic disorder, characterized by a cluster of cardiovascular risk factors (e.g., obesity, insulin resistance, and dyslipidemia). However, phenotypes of PCOS are often hardly separable and are defined by very heterogeneous clinical manifestations. Hence, when analysing the adherence to a dietary pattern, an anthropometric description mainly based on BMI, although divided into study group and controls, is too vague. During the recruitment phase, seemingly blood was drawn to define the biochemical hyperandrogenism values, but testosterone levels of the cohort are not described anywhere in the paper and should be added. The authors should include PCOS hormonal routine examination values (SHBG, LH, FSH, AMH, Cortisol, Testosteron, fTestosteron, 17ß-estradiol, Androstenedione, Progesteron), glycemic control values (fasting glucose, fasting insulin, HbA1c, HOMA index), lipid profile (HDL-c, LDL-c, Trigliceride), and cardio vascular risk markers (leptin, C-peptide, Hs-CRP, IL-6, IL-8, IL-18, TNF-alfa, MCP-1), along with the analysis methods.
My specific comments are provided below.
Line 45: “autoimmune disease” should not be stated or at least clarified, as attempts to identify an autoimmune cause in PCOS have been relatively uninformative. PMCID: PMC7417878
Line 67-69: The intention of this sentence is unclear. The introduction focuses particularly on the carbohydrate ratio in the different diets and how these low carbohydrates diet interventions are rarely compared to the benefits deriving from a Mediterranean diet. Please consider reformulating the sentences as it is not clear how the lack of comparison with the MED diet is in related to the DASH-derived benefits.
The authors should consider further explaining strengths and inherent limitations in applying these specific tools (dietary indices) to populations with a different dietary pattern for which they were designed.
Line 66: Within the context of interventional studies, successful attempts focusing on the long-term effects of the MED on incident CVD in men and women at high cardiovascular risk in Spain have been undertaken and could be discussed. For example PMID: 24829485.
Line 58: change to fiber.
Nothing more is mentioned though out the article regarding the relevance of fibres within the diet indices. Considering the most recent results, the interaction between fibers and microbiome and the effects on systemic inflammation should be mentioned. PMCID: PMC6940392
Line 124: Consider moving the entire description of the dietary indices to the introduction. Also, adding specific references for each diet, as the cited reference study (45) focuses mainly on aspects of DASH diet.
Line 129: Relationship between consumption and score need to be clarified (lowest consumption/highest score)
Line 133-134: Fix grammar
Line 140: For better readability, consider adding a flow diagram from patient recruitment process to the final analysis.
Line 161: “women with higher adherence to any of these 160 dietary scores had a higher age, physical activity and caffeine, carbohydrates and ω-3 fatty acids 161 intake, but lower BMI.” What is the significance of these results to the outcome of the study? Please mention in discussion section.
Since the results are not quite robust and somewhat inconclusive, and the reproducibility may be questioned because of insufficient evidence, the authours should consider expanding the discussion section based on the research questions expressed in the introduction and the results at hand (analysing the scores in the different quartiles based on the pattern composition of the mentioned diets for carbohydrates, monounsaturated fatty acids, and ω-3 polyunsaturated fatty acid, fibers) in order to support their hypotheisis and findings.
209: Fix wording: “join together the same syndrome”
210-211: The authors state in the intro “It may be due to diet may be more beneficial in these women than other phenotypes or controls”. This is a strong statement and misleading, considering the small sample size of the “H+O”subgroup and numerous other studies proving otherwise. What is the significance of this result based on the food characteristic of the specific diet scores? Since hormonal data are lacking, the authors should infer based on the material at hand: what are the differences based on the food scores and questionnaires between the lower scoring quartiles (Q2, Q3), compared to Q4?
211: fix grammer
212: This statement needs to be justified: in the table 1, quartiles describing BMI do not exactly define the associated PCOS phenotype; and furthermore the highest upper limit within the Q1 (reference) for AHEI-2010 is 30.9 which is not a state of obesity. The authors should consider describing with tables the exact characteristics of the groups.
226: ( Lizneva et al. 2016) : fix citation format
229-232: Fix wording, the intention of the sentence is not clear.
236: correct “(“
237: verb is missing in the sentence
239: correct grammer: “can better the risk”
253: correct grammer : “along the last decades”
253-254: fix grammer: the subject of the sentence is missing; it con not be inferred.
255-257: Fix wording, the intention of the sentence is not clear.
259-262: Fix wording: too many missing prepositions
282: Fix wording: missing prepositions
Author Response
Comments from Reviewer 4
Thank you for your very careful review of our paper, and for the comments, corrections and suggestions that ensued.
Comment 4-1: “The first main issue of this manuscript is the sample size that was generally small, and further divided into multiple subgroups. While one rule of thumb may be to take a sample of 10 times more than the no. of questions for 2 major study groups, I suggest the authors to report the calculations of sample size in the method section which allowed for further subgrouping while maintaining a sufficiently high study power, based also on prevalence studies of predominant PCOS phenotypes among the overall population”
Response 4-1: We agree with this comment. Therefore, Sample size was calculated based on other objective: to estimate mean difference of protein C, protein S and antithrombin III values between PCOS and controls in our previous work: “For an alpha error of 0.05 and 80% statistical power, a total of 126 individuals were needed in each group” Sanchez-Ferrer 2019. PMID: 31023609
Reference :
Sánchez-Ferrer ML, Prieto-Sánchez MT, Corbalán-Biyang S, Mendiola J, Adoamnei E, Hernández-Peñalver AI, Carmona-Barnosi A, Salido-Fiérrez EJ, Torres-Cantero AM. Are there differences in basal thrombophilias and C-reactive protein between women with or without PCOS? Reprod Biomed Online. 2019 Jun;38(6):1018-1026. doi: 10.1016/j.rbmo.2019.01.013. PMID: 31023609
Comment 4-2: “The second main issue is the lack of an in depth description of the hormonal and inflammatory values of the study groups. As mentioned in the introduction, PCOS is a lifelong reproductive and metabolic disorder, characterized by a cluster of cardiovascular risk factors (e.g., obesity, insulin resistance, and dyslipidemia). However, phenotypes of PCOS are often hardly separable and are defined by very heterogeneous clinical manifestations. Hence, when analysing the adherence to a dietary pattern, an anthropometric description mainly based on BMI, although divided into study group and controls, is too vague. During the recruitment phase, seemingly blood was drawn to define the biochemical hyperandrogenism values, but testosterone levels of the cohort are not described anywhere in the paper and should be added. The authors should include PCOS hormonal routine examination values (SHBG, LH, FSH, AMH, Cortisol, Testosteron, fTestosteron, 17ß-estradiol, Androstenedione, Progesteron), glycemic control values (fasting glucose, fasting insulin, HbA1c, HOMA index), lipid profile (HDL-c, LDL-c, Trigliceride), and cardio vascular risk markers (leptin, C-peptide, Hs-CRP, IL-6, IL-8, IL-18, TNF-alfa, MCP-1), along with the analysis methods.
Response 4-2: We published these data in a previous work. We have added the reference in methods in we have emphasized in results these data using the reference for not coping it. PMID: 31023609. However, we added hormonals and metabolic values in Table 1.
Line 199: “Table 1 shows demographic characteristics, metabolic parameters, hormonals determinations and nutrient intake across quartiles of adherence to healthful dietary scores for the study sample.”
Line 203: “Differences in demographic characteristics, metabolic parameters, and hormonals determinations among PCOS’ cases and controls have been published in a previous work [57]
Reference : Sánchez-Ferrer ML, Prieto-Sánchez MT, Corbalán-Biyang S, Mendiola J, Adoamnei E, Hernández-Peñalver AI, Carmona-Barnosi A, Salido-Fiérrez EJ, Torres-Cantero AM. Are there differences in basal thrombophilias and C-reactive protein between women with or without PCOS? Reprod Biomed Online. 2019 Jun;38(6):1018-1026. doi: 10.1016/j.rbmo.2019.01.013. PMID: 31023609
Comment 4-3: “Line 45: “autoimmune disease” should not be stated or at least clarified, as attempts to identify an autoimmune cause in PCOS have been relatively uninformative. PMCID: PMC7417878
Response 4-3: We refer to autoimmune thyroiditis. We eliminate this part of the quote of Patel 2018 (2) because as the reviewer 4 point, it may be confusing
Comment 4-4: “Line 67-69: The intention of this sentence is unclear. The introduction focuses particularly on the carbohydrate ratio in the different diets and how these low carbohydrates diet interventions are rarely compared to the benefits deriving from a Mediterranean diet. Please consider reformulating the sentences as it is not clear how the lack of comparison with the MED diet is in related to the DASH-derived benefits.
Response 4-4: Agree. We have reformulated the sentence: “Indeed, low carbohydrates diet interventions do not often compare with other frequent healthy dietary patterns such as Mediterranean or DASH diet as the control group”
Comment 4-4: The authors should consider further explaining strengths and inherent limitations in applying these specific tools (dietary indices) to populations with a different dietary pattern for which they were designed.
Response 4-4: We have incorporated the idea of the reviewer in the part of the discussion about limitation and strengths, line 317-321: “Second, we applied dietary indices which they have not been created specifically for Spanish population, except rMED index created for Spanish cohort of European Prospective Investigation into Cancer. This may limit external validity of this specific tool; however dietary indices provide a comparable gold standard and the indices that we have used are employed worldwide”.
Comment 4-5: “Line 66: Within the context of interventional studies, successful attempts focusing on the long-term effects of the MED on incident CVD in men and women at high cardiovascular risk in Spain have been undertaken and could be discussed. For example, PMID: 24829485”.
Response 4-5: We appreciate the reference and suggestion with regard to the incorporation of the following sentence, line 312-315: “For example, the PREDIMED study is one of the highest nutritional interventions, carried out in Spain. PREDIMEN has shown a 30% CVD risk reduction with the Mediterranean diet, which is of similar magnitude to that reported in the statin trials, although it is obtained at no cost for the health system ”
Comment 4-6: “Line 58: change to fiber.”
Response 4-6: Agree. We have been corrected
Comment 4-7: “Nothing more is mentioned though out the article regarding the relevance of fibres within the diet indices. Considering the most recent results, the interaction between fibers and microbiome and the effects on systemic inflammation should be mentioned. PMCID: PMC6940392”
Response 4-7: On the other hand, there is growing literature about the relation between diet, microbiome and gut-brain axis. It is clear the relation with obesity, digestive and endocrine and some mental disorders. It is not so well known the relation with cardiovascular disease and reproductive health but, as the reviewer mentions, it is an important point to consider. On the other hand, we did not include fiber as covariable because of statistical criteria and there were collinearity with carbohydrates ,that explained better the relation between dietary indices and PCOS . Authors agree with the reviewer it is important consider and we added a paragraph in discussion section about fiber, microbiome and female reproductive system
Line 305: “Another point to consider it is the importance of fiber on PCOS because of capacity of acting on gut microbiota as prebiotic. An increment of the gut permeability, the reduction of biodiversity and a growing endotoxemia by lipopolysaccharide of gram negative bacteria promote higher absorption of energy, active the immune system and inflammation, increase hyperandrogenism and insulin resistance (He and Li 2020). Gut microbiota has been implicated to play a critical role in metabolic diseases, such as PCOS, and may modulate the secretion of mediators of the brain–gut axis(Liu et al. 2017). However, the research is still in its early stage (Lindheim et al. 2017; Sanchez et al. 2020; Torres et al. 2018; Zhao et al. 2020). Fiber and microbiome may be a confounding factor in the present study. However, fiber did not accomplish statistical criteria for including in our analyses, whereas carbohydrates did. Also, fiber and carbohydrates intakes presented a high collinearity, for including both.”
Comment 4-8: “Line 124: Consider moving the entire description of the dietary indices to the introduction. Also, adding specific references for each diet, as the cited reference study (45) focuses mainly on aspects of DASH diet".
Response 4-8: We have rewritten all this part
Line 149: “The AHEI was developed to Teresa Fung and co-workers to quantify adherence to US federal dietary guidance of 1992, with a higher score reflecting better quality and adherence. The score ranges from 0 to 87.5 . This score evaluates 9 components such as trans fats, protein sources, polyunsaturated: monounsaturated ratio and cereal fiber. Table 1 describes the dietary indeces in detail. AHEI-2010 is the AHEI’s version for evaluating chronic diseases. AHEI and AHEI-2010 establish specific reference values of servings per day or grams per day for each food component and sum 10 if the subject reaches this amount. AHEI-2010 was designed in 2012 by Chiuve and colleagues based on update literature to study the relation between food intake and chronic. The AHEI-2010 scores 11 components for a total of 110 points, including whole grains intake acconding gender, legumes and nut, red:processed meat ratio, Sugar sweetened beverages and fruit juices, sodium and polyunsaturated fats [52]. The overall scoring range is 0 to 80 for AHEI and 0 to 110 for AHEI-2010. However, DASH, rMED and aMED establish the scoring criteria using quintiles, terciles and the median intake of the study sample, respectively. DASH was developed for controlling blood pressure [53] but, nowadays, is useful for obesity, diabetes, metabolic syndrome and cardiovascular disease. The DASH dietary pattern is rich in fruit, vegetables and low-fat dairy products. rMED and aMED indices define Mediterranean Diet and they are versions of the original Mediterranean Diet Score [54,55]. aMED considers red and processed meat and establishes ratio of mono/polyunsaturated fats [49], while rMED evaluates dairy products, only uses olive oil as the primary fat source, evaluates in one item all types of meat and is more specific for Spanish population [56]. The total score is 9 and 18, respectively. In all of these indices, higher intakes of healthy food items such as fruits, vegetables, whole grains, nuts and legumes add higher scores whereas higher intake of trans fats, meat, saturated fats, sodium and alcohol are associated with less score”
Comment 4-9: “Line 129: Relationship between consumption and score need to be clarified (lowest consumption/highest score)
Response 4-9: un mismo índice tiene algunos item sque van
highest consumption/highest score (consumo de verduras, frutas… lo que es positivo comer más)
lowest consumption/highest score (consumo de grasas saturadas, carne roja, alcohol… lo que es negativo comer más)
voy a especificarlo en una tabla seguramente
Comment 4-10 “Line 133-134: Fix grammar”
Response 4- 10: We have change the link and the “s” of version , line 152-153: “rMED and aMED indices define Mediterranean Diet and they are versions of the original MDS [50,51].”
Comment 4-11: “Line 140: For better readability, consider adding a flow diagram from patient recruitment process to the final analysis.”
Response 4-11: The authors agree, we have added the next figure of the flow chart
|
Figure 1. Flow chart of inclusion and exclusion criteria of PCOS’ case-control study of Shoutheast of Spain from 2014 to 2016 |
Comment 4-12: “Line 161: “women with higher adherence to any of these 160 dietary scores had a higher age, physical activity and caffeine, carbohydrates and ω-3 fatty acids 161 intake, but lower BMI.” What is the significance of these results to the outcome of the study? Please mention in discussion section”
Response 4-12:
Comment 4-13: Since the results are not quite robust and somewhat inconclusive, and the reproducibility may be questioned because of insufficient evidence, the authours should consider expanding the discussion section based on the research questions expressed in the introduction and the results at hand (analysing the scores in the different quartiles based on the pattern composition of the mentioned diets for carbohydrates, monounsaturated fatty acids, and ω-3 polyunsaturated fatty acid, fibers) in order to support their hypotheisis and findings.
Response 4-13:
Comment 4-14: 209: Fix wording: “join together the same syndrome”
Response 4-14: “In fact, clinical manifestations of PCOS are notably different among PCOS phenotypes, such as presenting androgenism or not, although all of them take part of the same syndrome [67]”
Comment 4-15: 210-211: The authors state in the intro “It may be due to diet may be more beneficial in these women than other phenotypes or controls”. This is a strong statement and misleading, considering the small sample size of the “H+O”subgroup and numerous other studies proving otherwise. What is the significance of this result based on the food characteristic of the specific diet scores? Since hormonal data are lacking, the authors should infer based on the material at hand: what are the differences based on the food scores and questionnaires between the lower scoring quartiles (Q2, Q3), compared to Q4?
Response 4-15: We agree. We have rested importance: “One possibility which may explain of this association is that women with “H+O” phenotype presented higher prevalence of overweight and obesity (52.9%) compared to controls (33.1%) and other PCOS phenotypes (33.9%), but we need further studies with higher sample size.”
Comment 16: “211: fix grammar”
Response 4-16: The grammatical error “It may be due to diet may be more beneficial in these women” have change to: “we think it is interesting explore differences between phenotypes due to diet may be more beneficial for women with a particular phenotype than for others”. We have changed the idea because we think the reviewer have reason and we must take off importance to these result due to the lack of statistical power.
Comment 4-17: “212: This statement needs to be justified: in the table 1, quartiles describing BMI do not exactly define the associated PCOS phenotype; and furthermore the highest upper limit within the Q1 (reference) for AHEI-2010 is 30.9 which is not a state of obesity. The authors should consider describing with tables the exact characteristics of the groups.”
Response 4-17:
Comment 4-18:
“226: ( Lizneva et al. 2016) : fix citation format
Response 4-18:
Comment 4-19: “229-232: Fix wording, the intention of the sentence is not clear”.
Respond 4-19: We have changed to this sentence “The other different item is sugar-sweetened beverages and fruit juice. AHEI-2010 scored 0 points if they drink one or more servings per day and the maximum score when subjects do not drink any sugar-sweetened beverages and fruit juice. DASH score has an item for sweetened beverages too”
Comment 4-20: “236: correct “(“
237: verb is missing in the sentence
239: correct grammer: “can better the risk”
253: correct grammer : “along the last decades”
253-254: fix grammer: the subject of the sentence is missing; it con not be inferred.
255-257: Fix wording, the intention of the sentence is not clear.
259-262: Fix wording: too many missing prepositions”
Response 4-20: Thank you for pointing these mistakes in detail. All spelling and grammatical errors pointed out by the reviewer have been corrected.

Round 2
Reviewer 4 Report
The new adjustments by the Authors add readability, prospective significance and logical reasoning to the trending results.
Some remaining issues:
Your Response 4-2 in the Author's Notes is not present in the paper: "We published these data in a previous work. We have added the reference in methods in we have emphasized in results these data using the reference for not coping it. PMID: 31023609. However, we added hormonals and metabolic values in Table 1. (missing)"
Line 91-92: As mentioned sample size is based on previous study (reference 37). This detail should be stated separately and rather in the statistical analysis section.
2 tables are described with the same number:
Line 150 "Table 1" for dietary indices
Line 201 “Table 1” for demographic characteristics and nutrient intake
Line 201: the cited “metabolic parameters, hormonal determinations” data/table is actually not present
Line 292: Consider moving this to line 288, before the explanatory arugments.
Please correct the numerous typos.